# Urinary 15-F_2t_-Isoprostane Concentrations in Dogs with Liver Disease

**DOI:** 10.3390/vetsci10020082

**Published:** 2023-01-21

**Authors:** Robert Kyle Phillips, Jörg M. Steiner, Jan S. Suchodolski, Jonathan A. Lidbury

**Affiliations:** Gastrointestinal Laboratory, Department of Small Animal Clinical Sciences, School of Veterinary Medicine and Biomedical Sciences, Texas A&M University, College Station, TX 77843-4474, USA

**Keywords:** oxidative stress, urinary isoprostane, liver disease, dog, canine hepatopathy, congenital portosystemic shunt

## Abstract

**Simple Summary:**

Similar to humans, dogs are affected by a variety of liver diseases with various causes and presentations. Recently, growing interest has focused on measuring markers of oxidative stress in patients with liver disease; however, no study to date has compared markers of oxidative stress between dogs with different liver diseases. This study aimed to evaluate markers of oxidative stress in dogs with different types of liver disease by measuring a compound known as 15-F_2t_-isoprostane in their urine. Among dogs with one of three different types of liver diseases, only those with a congenital portosystemic shunt (CPSS) had a significant increase in this oxidative stress marker when compared with a group of healthy control dogs.

**Abstract:**

Isoprostanes are stable end products of lipid peroxidation that can be used as markers of oxidative stress. It was previously reported that a cohort of dogs with various liver diseases had increased urinary isoprostane concentrations compared to healthy control (HC) dogs. The aim of this study was to measure and report urinary isoprostane concentrations in dogs with different types of liver diseases. Urine was collected from 21 HC dogs and from 40 dogs with liver disease, including 25 with chronic hepatitis (CH), 7 with steroid hepatopathy (SH), and 8 with a congenital portosystemic shunt (CPSS). In this prospective, observational study, urinary 15-F_2t_-isoprostane (F_2_-IsoP) concentrations were measured by liquid chromatography/mass spectrometry and normalized to urinary creatinine concentrations. Concentrations were compared between groups using a Kruskal–Wallis test followed by Dunn’s multiple comparisons tests. Significance was set at *p* < 0.05. The median (range) urinary F_2_-IsoP to creatinine ratios (ng/mg UCr) were 3.6 (2.2–12.4) for HC dogs, 5.7 (2.4–11.3) for dogs with CH, 4.8 (2.4–8.6) for dogs with SH, and 12.5 (2.9–22.9) for dogs with CPSS. CPSS dogs had significantly higher urinary F_2_-IsoP concentrations than HC dogs (*p* = 0.004), suggesting increased oxidative stress among this cohort.

## 1. Introduction

Oxidative stress occurs when the concentration of pro-oxidant molecules and the supply of antioxidant defenses become imbalanced [1]. Left unchecked, reactive oxygen species (ROS) may nonspecifically react with biological molecules, resulting in structural and functional damage to cells. Increased oxidative damage may be the result of an increase in the production of ROS, a decrease in the availability of antioxidant molecules or enzymes, or both [2].

Lipids, a primary target for ROS, can produce a multitude of different products of free radical chemistry [3]. The measurement of F_2_-isoprostanes (F_2_-IsoP) is considered the most reliable biomarker of lipid peroxidation in vivo [4]. F_2_-IsoPs are generated from the nonenzymatic, free-radical-induced oxidation of arachidonic acid and have been used to implicate the role of oxidative damage in the pathogenesis of numerous disorders such as cystic fibrosis, chronic renal insufficiency, congestive heart failure, diabetes, and pulmonary hypertension [5,6].

Focus is growing on the mechanism by which oxidative stress may be involved in the development of various liver disorders in humans, including alcoholic liver disease [7,8,9], nonalcoholic fatty liver disease [10,11,12], hepatitis C [13,14,15], and hepatic encephalopathy [16,17,18]. However, few studies have investigated the interplay between oxidative stress and liver disease in dogs [19]. One recent paper described the biomarkers of oxidative stress, including F_2_-IsoP, in dogs with different liver diseases undergoing a liver biopsy [20]. Urinary F_2_-IsoP concentrations were elevated in the single heterogenous liver disease population compared with those in healthy control dogs; however, comparisons across different liver disease cohorts were not evaluated.

Therefore, the objective of this study was to measure urinary F_2_-IsoP concentrations (as a marker of lipid peroxidation) in dogs with different types of liver diseases. We hypothesized that dogs with certain types of liver diseases would have increased urinary F_2_-IsoP concentrations when compared with those of healthy controls.

## 2. Materials and Methods

### 2.1. Animals

Client-owned dogs of any age and sex, weighing greater than 3 kg bodyweight, and presenting to the Texas A&M University Veterinary Medical Teaching Hospital (VMTH) or Gulf Coast Veterinary Specialists with a definitive diagnosis of a specific liver disease, arrived at either histologically or through diagnostic imaging, between 1 May 2019, and 30 November 2021, were prospectively enrolled into this study. In addition, during the spring and fall semesters of 2021, an additional 21 dogs owned by the faculty and staff of the VMTH were recruited as healthy controls (HC). The health of these dogs was determined by the absence of clinical signs based on owner questionnaire; physical examination; and the lack of clinically relevant abnormalities on a complete blood count, urinalysis, and serum biochemistry profile. The animal use protocol for this study was approved by the Institutional Animal Care and Use Committee of Texas A&M University (IACUC 2017-0351 and IACUC 2020-0248). Informed owner consent was obtained prior to study enrollment for all dogs.

Where clinically indicated for diseased dogs, 4 to 6 laparoscopic liver biopsy specimens were collected and submitted for routine histopathology (including rhodanine staining for copper) and for tissue copper quantification by flame atomic absorption spectrometry at the Veterinary Diagnostic Laboratory at Colorado State University (Fort Collins, CO, USA). The diagnoses of hepatic disease, based on micromorphological diagnosis by a board-certified veterinary anatomic pathologists, routine bloodwork, and diagnostic imaging (i.e., CT angiography and ultrasound interpreted by a board-certified veterinary radiologist), when available, were used to identify dogs belonging to the following three liver disease groups: chronic hepatitis (CH), steroid hepatopathy (SH), or congenital portosystemic shunt (CPSS). Basic demographic information (i.e., breed, sex, age, and weight) as well as a current medication list (including whether a dog was taking any antioxidant supplements) were collected at time of enrollment.

### 2.2. Sample Collection and Processing

Urine samples were collected by either free catch or cystocentesis. After centrifugation (400 *rcf* for 5 min at room temperature), 1 mL aliquots were made and immediately frozen at −80 °C until analysis as a single batch. F_2_-IsoP (specifically the 15-F_2t_-isoprostane isomer) was quantified by liquid chromatography/mass spectroscopy at the Eicosanoid Core Laboratory at Vanderbilt University (Nashville, TN, USA), following methods based on their published protocol [21] and outlined in Appendix A. Final F_2_-IsoP concentrations were normalized against urinary creatinine (UCr) concentrations and reported in units ng/mg UCr.

### 2.3. Statistical Analysis

A power analysis was performed based on the mean urinary F_2_-IsoP concentrations of healthy dogs reported in an earlier study [22]. In order to achieve 90% power while detecting a conservative fourfold change in F_2_-IsoP concentrations among groups, the target enrollment was calculated to be 28 dogs in total (i.e., 7 dogs per group).

Shapiro–Wilk tests and visual inspection of q-q plots characterized the data as having a non-normal distribution. F_2_-IsoP concentrations, age, and bodyweight among groups were compared using Kruskal–Wallis tests followed by Dunn’s multiple comparisons tests, as needed. Wilcoxon exact tests were used to compare the effect of antioxidant supplementation (AOS) status, bodyweight, sex, and age on F_2_-IsoP concentrations within groups. Spearman’s correlation was used to assess the relationship between hepatic copper concentration (where measured) and urinary F_2_-IsoP concentrations. Results were calculated using a commercially available software package (Prism v.8.3.0 for Windows, GraphPad Software, San Diego, CA, USA). Significance was set at *p* < 0.05.

## 3. Results

### 3.1. Study Population

A total of 61 dogs were enrolled into this study: 21 HC dogs, 25 CH dogs, 7 SH dogs, and 8 CPSS dogs. The basic descriptive characteristics of healthy control dogs versus liver disease dogs are summarized in Table 1. The study population included 19 mixed-breed dogs. The most commonly represented purebred dog breeds (count) included miniature schnauzers (6), Labrador retrievers (5), Australian shepherds (3), Rhodesian ridgebacks (3), Cairn terriers (2), golden retrievers (2), Maltese (2), pugs (2), and standard poodles (2). The remaining purebred dogs included a single dog of each of the following breeds: black and tan coonhound, black Russian terrier, blue heeler, bluetick coonhound, boxer, Brittany spaniel, Chihuahua, dachshund, German shepherd, Great Dane, Old English sheepdog, rat terrier, Siberian husky, Weimaraner, and Yorkshire terrier. A visual representation of the breed makeup by liver disease group is provided in Appendix A.

CH and SH dogs were significantly older than those in the HC group (*p* = 0.001, *p* = 0.015) or the CPSS group (*p* < 0.001, *p* = 0.001). Dogs in the CPSS group were significantly lighter than dogs in the HC group (*p* = 0.002). All other comparisons between groups were not significantly different (*p* > 0.05). Sex distribution was not associated with group (*p* = 0.487). These results are graphically presented in Appendix A.

### 3.2. Urinary 15-F_2t_-Isoprostane Concentrations

The median [range] F_2_-IsoP/UCr concentrations for each group are summarized in Appendix A. A significant increase in F_2_-IsoP concentration was observed when comparing the combined group (LD) of all dogs with liver disease (median: 6.2 ng/mg UCr [2.4–22.9]) with HC dogs (median: 3.6 ng/mg UCr [2.2–12.4], *p* = 0.001, Appendix A). When separately considering each liver disease cohort (Figure 1), F_2_-IsoP concentrations were significantly higher in CPSS dogs (median: 12.5 ng/mg UCr [2.9–22.9]) than in HC dogs (*p* < 0.001). F_2_-IsoP concentrations in dogs with CH (5.7 ng/mg UCr [2.4–11.3]) and in dogs with SH (4.8 ng/mg UCr [2.4–8.6]) trended higher than those in HC dogs; however, these increases were not statistically significant (*p* = 0.082 and *p* > 0.999, respectively). Appendix A lists the *P*-values for all comparisons among groups. The F_2_-IsoP/UCr concentrations for individual dogs are provided in Appendix A.

The tissue copper quantification results for CH dogs greatly varied, ranging from 12 ppm to 3108 ppm (median: 407 ppm, Appendix A). No correlation appeared to exist between F_2_-IsoP concentrations and the amount of copper measured in the liver biopsies from CH dogs (Spearman r = −0.114, *p* = 0.596, Figure 2).

Within each group of dogs, F_2_-IsoP concentrations were not significantly different for those receiving antioxidant supplementation (i.e., administration of S-adenosylmethionine, milk thistle extract, or vitamin E). Table 2 details the individual *P*-values for each within-group comparison.

## 4. Discussion

The results of this prospective study demonstrated a significant difference in the concentration of urinary F_2_-IsoP among HC dogs and dogs with liver disease. In particular, we found that dogs with CPSS had significantly higher urinary F_2_-IsoP concentrations than HC dogs.

The heterogeneity of F_2_-IsoP concentrations among liver disease groups observed in our study recapitulates findings in a meta-analysis of F_2_-IsoP concentrations in human studies across multiple pathologies. In particular, marked differences in the degree of F_2_-IsoP increase in patients versus healthy controls were reported depending on the specific liver disease etiology being evaluated (i.e., alcoholic liver disease > hepatitis C > nonalcoholic fatty liver disease > autoimmune hepatitis > hepatic cirrhosis) [6]. Similarly, in the current study, it appeared that of the three liver disease cohorts investigated, CPSS had a greater effect on a dog’s oxidative stress (as estimated by the measurement of urinary F_2_-IsoP) than either chronic hepatitis or steroid hepatopathy.

The significant elevation in F_2_-IsoP concentrations in the supergroup of all dogs with liver disease (LD) as compared with HC dogs mirrors the findings presented in the previous study by Barry-Heffernan et al. [20]. The present study, which included more CH dogs (25 vs. 21 dogs with “inflammatory” liver disease) and more CPSS dogs (8 vs. 2 dogs with “vascular” liver disease), was able to make comparisons among individual liver disease cohorts, wherein only the group of CPSS dogs was significantly different. While the CH dogs in our study had F_2_-IsoP concentrations trending higher than the HC dogs, this increase was not statistically significant at the *p* < 0.05 level. Comparing studies, our group of HC dogs had a slightly higher median F_2_-IsoP concentration versus the group of control dogs in the previous study (3.6 ng/mg UCr and 2.98 ng/mg UCr, respectively). This difference may have been one contributing factor to how their largely “inflammatory” disease-based liver disease group (21 out of 34 dogs) achieved a significant difference from their control group while our group of CH dogs did not.

Elevated plasma ammonia concentrations are associated with the development of hepatic encephalopathy (HE) in dogs [23] and people [24], but multiple other factors, including oxidative stress, may play a synergistic role with ammonia in the pathogenesis of HE [17,25]. In one study of rats with experimentally created portocaval shunts, both ammonia levels and multiple markers of oxidative stress were significantly increased six weeks after surgery as compared with those of sham-operated rats [26]. Another study in humans found that a single serum marker of oxidative stress was able to distinguish between cirrhotic patients with and without mild HE signs [27]. Oxidative stress is generally considered an important factor in the pathogenesis of chronic liver disease in dogs [28], but the role it plays in the pathogenesis of HE requires greater investigation [29]. While we did not assess signs of HE in our cohort of CPSS dogs, it would be interesting in future studies to determine if there is a correlation between urinary F_2_-IsoP concentrations and clinical severity of HE. It would also be valuable to see if increases in urinary F_2_-IsoP concentrations persist post-shunt attenuation.

Demographic characteristics (notably bodyweight or age) were significantly different across groups, but this may largely be explained by features of the underlying disease process. That is, the bodyweight of unhealthy dogs may be diminished as a reflection of their failure to thrive. Chronic conditions such as CH and SH generally affect dogs later in life. CPSS, a disorder present from birth, is frequently diagnosed earlier in life and often in smaller breed patients. An analysis of urinary F_2_-IsoP concentrations by individual demographic attributes showed no significant differences at *p* < 0.05; however, male HC and male CPSS dogs did have near-significant decreases in urinary F_2_-IsoP concentrations when compared with females within the same cohort (*p* = 0.051 and *p* = 0.071, respectively). Interestingly, this observation runs contrary to the generally established paradigm in humans and rats that male subjects tend to have greater oxidative stress than female subjects [30,31].

No clear evidence for a difference in urinary F_2_-IsoP concentration was appreciated among the dogs in this study who were on supplementation with hepatic cytoprotective agents, including antioxidants, versus dogs who were not supplemented. Indeed, the efficacy of antioxidant therapy as assessed by biomarkers of oxidative stress remains a highly controversial topic. While many researchers remain optimistic about the utility of antioxidant supplementation (AOS) [19,32,33], others have reported a more guarded outlook [1,34,35]. To maximize success, the current AOS philosophy calls for increased precision in the field of antioxidant pharmacology, purposefully selecting AOS agents and dosages that are geared toward effectively targeting specific ROS within specific tissues in vivo as compared with a more imprecise, monotherapy approach [36]. Biomarkers of oxidative stress, such as F_2_-IsoP, may yet help clinicians achieve this targeted approach.

This study had several limitations. First, for some of our liver disease groups (namely SH dogs and CPSS dogs), we achieved the bare minimum sample size based on a power analysis. A larger study that also includes numerous enrollees with other liver disease etiologies (e.g., acute liver injury, hepatic neoplasia, nonhepatic disease, and copper-associated chronic hepatitis) would serve to broaden the scope of the findings reported here. Second, the HC enrolled in this study were not age- or breed-matched. However, this would have been challenging, as the age/breed distribution for each type of liver disease was different. Third, the single urine sample we collected from each dog only allowed for the quantification of oxidative stress levels at a single point in time. One study looking at urinary F_2_-IsoP concentrations in healthy humans (n = 13) found the day-to-day variation to be 42% over 10 consecutive days [37]. For any future studies, collecting multiple samples at different timepoints could serve to mitigate this variability and, with a longitudinal design, could be structured to assess changes in oxidative stress over the course of treatment. Furthermore, the exact method of urine collection in this study (by free-catch or by cystocentesis) was neither standardized across patients nor consistently recorded in the patients’ medical records. While we do not think there is any effect on urinary F_2_-IsoP concentration based on urine collection method, to be fully confident, this variable should be purposefully investigated in a future study. Fourth, as we did not control for medication use, the effect of medical treatment on urinary F_2_-IsoP concentrations, especially S-adenosylmethionine, milk thistle extract, or vitamin E supplementation, cannot be easily teased out. While there was no significant difference in the urinary F_2_-IsoP concentrations between dogs on or off AOS within groups, it may be the case that AOS treatment was disproportionately reserved for patients with more severe disease, thus obscuring our ability to judge how effective AOS is at ameliorating oxidative stress. We did re-analyze our data after excluding all dogs receiving AOS (data not shown) only to arrive at the same overall conclusion: only the comparison between CPSS dogs and HC dogs showed a significant difference in urinary F_2_-IsoP concentration (*p* = 0.003).

## 5. Conclusions

We conclude that dogs with CPSS had increased urinary F_2_-IsoP concentrations compared with HC dogs, suggesting increased oxidative stress. If these findings are confirmed in a larger group of dogs, this may prove useful in focusing in on which liver disease patients are most in need of antioxidant therapy.

## Figures and Tables

**Figure 1 vetsci-10-00082-f001:**
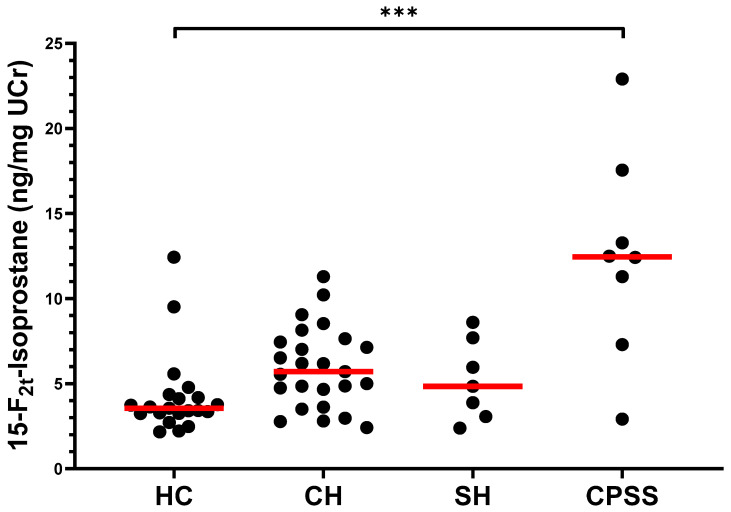
Concentrations of 15-F_2t_-isoprostane (ng/mg UCr) in the urine of healthy dogs and dogs with CH, SH, or CPSS. The median of each group is represented by a red line. Asterisks (***) represent a significant difference at *p* < 0.001. All other comparisons between groups were not significantly different at *p* = 0.05. Abbreviations: HC, healthy controls; CH, chronic hepatitis; SH, steroid hepatopathy; CPSS, congenital portosystemic shunt; UCr, urinary creatinine.

**Figure 2 vetsci-10-00082-f002:**
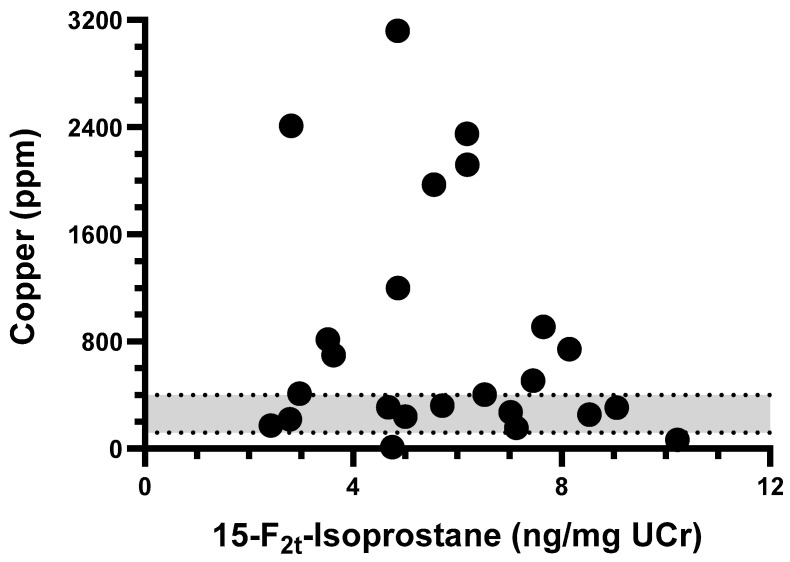
XY-plot of tissue copper quantification and urinary 15-F_2t_-Isoprostane concentration among dogs in chronic hepatitis group. No correlation was observed between variables (Spearman r = −0.114, *p* = 0.596). Gray area represents normal tissue copper concentration range (120–400 ppm).

**Table 1 vetsci-10-00082-t001:** Signalment, bodyweight, breed distribution, and antioxidant supplementation status of healthy control dogs and dogs with CH, SH, or CPSS. Age and bodyweights are reported as medians [ranges].

Variable	HC	CH	SH	CPSS
Number (n=)	21	25	7	8
Age (years)	4 [1.5–9]	8.6 [1–13]	9.4 [6–11]	2 [1–6]
Weight (kg)	29.5 [12.7–43.6]	21.8 [2.9–51.2]	17.6 [3.3–53.4]	6.2 [4.4–24.6]
Male (N/I)	11 (9/2)	11 (11/0)	3 (3/0)	2 (1/1)
Female (S/I)	10 (8/2)	14 (13/1)	4 (4/0)	6 (5/1)
Purebred/Mix	12/9	18/7	5/2	7/1
AOS, Yes/No	0/21	16/9	2/5	1/7

Abbreviations: HC, healthy controls; CH, chronic hepatitis; SH, steroid hepatopathy; CPSS, congenital portosystemic shunt; I, intact; N, neutered; S, spayed; AOS, antioxidant supplementation.

**Table 2 vetsci-10-00082-t002:** Comparisons of within-group effect of antioxidant supplementation status, bodyweight, sex, and age on median urinary 15-F_2t_-isoprostane concentrations (ng/mg UCr).

		HC				CH				SH				CPSS			
		*n*	Med.	Range	*p*-Value	*n*	Med.	Range	*p*-Value	*n*	Med.	Range	*p*-Value	*n*	Med.	Range	*p*-Value
AOS	Yes	0	n.d. ^a^	n.d. ^a^	n.d. ^a^	16	6.77	2.42–11.29	0.677	2	3.48	3.07–3.88	0.381	1	17.55	17.55–17.55	0.500
	No	21	3.56	2.17–12.43		9	5.55	2.81–9.06		5	5.96	2.39–8.61		7	12.42	2.92–22.90	
BW	<25 kg	7	3.63	3.24–5.58	0.443	10 ^b^	6.19	2.78–10.22	0.860 ^b^	5	3.88	2.39–8.61	0.381	8	12.49	7.3–22.9	n.d. ^c^
	≥25 kg	14	3.39	2.17–12.43		5 ^b^	5.55	2.81–8.53		2	6.83	5.96–7.70		0	n.d. ^c^	n.d. ^c^	
Sex	M	11	3.29	2.17–4.78	0.051	11	5.55	2.81–11.29	0.536	3	3.88	2.39–4.84	0.229	2	5.11	2.92–7.30	0.071
	F	10	3.76	2.48–12.43		14	5.95	2.42–9.06		4	6.83	3.07–8.61		6	12.89	11.29–22.90	
Age	<6 yr.	18	3.58	2.17–12.43	0.534	7	6.18	2.78–11.29	0.883	0	n.d. ^d^	n.d. ^d^	n.d. ^d^	7	12.49	2.92–22.9	1.000
	≥6 yr.	3	3.56	2.48–3.63		18	5.63	2.42–10.22		7	4.84	2.39–8.61		1	12.42	12.42–12.42	

Abbreviations: HC, healthy controls; CH, chronic hepatitis; SH, steroid hepatopathy; CPSS, congenital portosystemic shunt; AOS, antioxidant supplementation; BW, bodyweight; Med., median; M, male; F, female; yr., years; n.d., not determined; UCr, urinary creatinine. ^a^ Lack of any HC dogs on antioxidant supplementation. ^b^ Bodyweights not available for 10 CH dogs. ^c^ All CPSS dogs weighed under 25 kg. ^d^ No SH dogs were younger than 6 years old.

## Data Availability

The data presented in this study are available either within the article or as Appendix A.

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
