# Peer review of "Urinary 15-F2t-Isoprostane Concentrations in Dogs with Liver Disease"

_vetsci, 2023, doi:10.3390/vetsci10020082_

Round 1
Reviewer 1 Report
The paper is well organized and the research is well designed. The conclusions are balanced and supported by the results. I would only recommend the clarification in the passage lines 137-144. aIt is a bit difficult to follow what authors mean by "nor did any other comparison reach statistical significance" or why "A significant increase in F2-IsoP compared with HC dogs persisted even when all liver disease dogs were considered as a single, combined liver disease (LD) group (median: 6.185 ng/mg UCr [2.4 – 22.90], P = .01" if a sentence before they said no significant differences were observed.
Reviewer 2 Report
In this manuscript, the author reviewed the urinary concentration of 15F2t isoprostane in dogs with liver diseases. F2-IsoP is well known to be as an oxidative stress marker even in dogs.
- Urinary F2-IsoP concentrations in dogs with liver diseases were documented in the previous report [20]. It said that the urine F2-IsoP/cre elevated in the liver diseases compared with healthy control dogs. While the liver diseases were heterogenous in that study, most of them (n=21) were inflammatory diseases and just only two cases were vascular (portal vein hypoplasia). On the other hand, cPSS dogs had the high levels of F2-IsoP, but other liver disease such as chronic hepatitis dogs did not in this study.
1-1) How would you interpret the difference between these results? The authors should describe in the Discussion part.
1-2) The authors described “More granular comparisons across different liver disease cohorts were not evaluated due to limited sample size [20].” They collected 34 liver disease-cases, which is not so different compared to this study (n=40). The authors should describe other reason(s).
- In this study, a nonignorable number of cases were treated with anti-oxidant drugs. They documented it as the limitation of this study, but the reviewer thinks it is needed to focus on it harder.
2-1) The authors checked the possibility of confounding of AOS in the supplemental Table 1. Since this information is important, it is better to be posted in the body.
2-2) I would propose that authors should analyze using only AOS-nontreated dogs. If the results could be quite changed after excluding AOS-treated cases, please show the data in the body. If not, it is not necessary.
2-3) In the supplemental Table 1, some statistical analysis was performed although some groups were n=1. Please make it clear what kind of analysis method was used.
3) Do you have the time sequence data of some cases? It should be worthy to be added because the authors could evaluate whether the oxidative status/symptoms/severity/treatment are related to F2-IsoP.
4) The urine samples were collected by the methos of free catch or cystocentesis. That is just some unpublished data, but the reviewer has the experience that the collecting methods affects the levels of some lipid metabolites. Please check the concentration changes related to the collecting methods.
5) Table 2 and 3 should be omitted. These data could be understood in the text. And then, the supplemental Table 1 and Figure 7 should be posted in the body. These are negative results, but they are important and difficult to image based on only the text.
Reviewer 3 Report
15-F2t-isoprostane concentrations are usually detected using a liquid chromatographic mass spectrometer, and raw chromatographic mass spectrometry, chromatographic conditions, mass spectrometry conditions, linear relationships, precision, recovery, and stability data are provided. This paper needs to supplement the above experimental content. Please explain why a gas chromatograph mass spectrometer is used to detect the 15-F2t-isoprostane concentration.
Round 2
Reviewer 2 Report
Nothng to comment
Author Response
Thank you for reviewing our manuscript.
There were not any additional commentary/suggestions from this reviewer that required changes to the manuscript.
Reviewer 3 Report
This paper would have been of more solid scientific interest if the authors had been able to provide the detection limits, some of the original plots, standard curves, precision, and recoveries for the LC-MS of 15-F2t-isoprostane. The authors did not provide the original plots making it impossible for the reader to make a direct comparison and confirmation of the results. Thus, the scientific significance of this paper is reduced.
